# The Effect of Voriconazole on Tacrolimus in Kidney Transplantation Recipients: A Real-World Study

**DOI:** 10.3390/pharmaceutics14122739

**Published:** 2022-12-07

**Authors:** Yi-Chang Zhao, Chen-Lin Xiao, Jing-Jing Hou, Jia-Kai Li, Bi-Kui Zhang, Xu-Biao Xie, Chun-Hua Fang, Feng-Hua Peng, Indy Sandaradura, Miao Yan

**Affiliations:** 1Department of Pharmacy, The Second Xiangya Hospital of Central South University, Changsha 410011, China; 2International Research Center for Precision Medicine, Transformative Technology and Software Services, Changsha 410011, China; 3Department of Urological Organ Transplantation, The Second Xiangya Hospital of Central South University, Changsha 410011, China; 4School of Medicine, University of New South Wales, Sydney, NSW 2052, Australia; 5Centre for Infectious Diseases and Microbiology, Westmead Hospital, Sydney, NSW 2145, Australia

**Keywords:** tacrolimus, voriconazole, kidney transplantation, drug interaction

## Abstract

Tacrolimus is an immunosuppressant with a narrow therapeutic window. Tacrolimus exposure increased significantly during voriconazole co-therapy. The magnitude of this interaction is highly variable, but it is hard to predict quantitatively. We conducted a study on 91 kidney transplantation recipients with voriconazole co-therapy. Furthermore, 1701 tacrolimus concentration data were collected. Standard concentration adjusted by tacrolimus daily dose (C/D) and weight-adjusted standard concentration (CDW) increased to 6 times higher during voriconazole co-therapy. C/D and CDW increased with voriconazole concentration. Patients with the genotype of CYP3A5 *3/*3 and CYP2C19 *2/*2 or *2/*3 were more variable at the same voriconazole concentration level. The final prediction model could explain 54.27% of the variation in C/D and 51.11% of the variation in CDW. In conclusion, voriconazole was the main factor causing C/D and CDW variation, and the effect intensity should be quantitative by its concentration. Kidney transplant recipients with CYP3A5 genotype of *3/*3 and CYP2C19 genotype of *2/*2 and *2/*3 should be given more attention during voriconazole co-therapy. The prediction model established in this study may help to reduce the occurrence of rejection.

## 1. Introduction

For patients with end-stage kidney disease, transplantation is a preferred treatment option. Tacrolimus is a calcineurin inhibitor introduced in the 1990s. It is the cornerstone of most immunosuppressive regimens after solid organ transplantation [1,2,3]. Its use has revolutionized transplantation. Moreover, it has been associated with better graft survival, lower incidence of rejection, and improved drug tolerance with fewer side effects [4,5]. Its therapeutic window is narrow, and significant interindividual and intraindividual differences in pharmacokinetic parameters have been described [6,7,8,9]. An appropriate concentration of tacrolimus is necessary to avoid adverse reactions and kidney graft rejections [4,10,11]. Guidelines recommended using tacrolimus, mycophenolate mofetil, and glucocorticoid as a triple therapy regimen [12]. At the same time, invasive aspergillosis was frequently reported to occur after renal transplantation [13,14,15], which is associated with increased mortality. Voriconazole was the first-line treatment [15,16]. It is predominantly metabolized by cytochrome CYP450 isozymes CYP2C19, CYP2C9, and CYP3A4, among which CYP2C19 is the primary metabolic pathway [17,18,19,20]. Tacrolimus blood concentration could increase up to 10-fold with voriconazole co-therapy [21]. The influence of voriconazole on tacrolimus varies between individuals. The effect may be related to CYP3A4 and CYP3A5 gene polymorphisms [22]. During voriconazole co-therapy, the instruction book recommends reducing the tacrolimus dose to one-third [23]. Nevertheless, abundant studies suggested that the recommended dose adjustment may fail to maintain tacrolimus concentrations within the required therapeutic range [24,25,26,27]. Therefore, it is necessary to monitor tacrolimus trough concentration continuously and adjust the dose after kidney transplantation to avoid substantial changes in drug concentration. Despite this, dose adjustment protocols for tacrolimus in combination with voriconazole remain empirical. It is urgent to formulate individualized tacrolimus dosing regimens in this particular population. This study aimed to identify genetic and clinical predictors of the magnitude of drug and drug interaction (DDI) between tacrolimus and voriconazole and reduce rejection caused by drastic changes in tacrolimus.

## 2. Materials and Methods

### 2.1. Study Design and Data Collection

This study was conducted based on data from the Department of Kidney Transplantation of the Second Xiangya Hospital of Central South University and approved by the Ethics committee of our hospital ((2020) Ethical Review (CR) No. (077)). The clinical and research activities are consistent with the Principles of the Declaration of Istanbul outlined in the Declaration of Istanbul on Organ Trafficking and Transplant Tourism.

We collected clinical and laboratory data between January 2016 and December 2021. Inclusion criteria were the following: (1) after first kidney transplantation; (2) age ≥ 18; (3) patients in hospital; (4) treated with a triple immunosuppression therapy regimen of tacrolimus, mycophenolate mofetil, and glucocorticoids; (5) at least one voriconazole steady-state trough concentration available during concomitant use of voriconazole. Exclusion criteria were as follows: (1) patients with missing tacrolimus dosing and concentration data; (2) patients undergoing kidney replacement therapy such as hemodialysis and plasmapheresis; (3) with Lack of CYP3A5 genotype and CYP2C19 genotype information. The laboratory information system (LIS), hospital information system (HIS), and TDM system were used to retrieve data. Demographic information, voriconazole prescription data, and drug combination information were collected. The dose and concentration of voriconazole before voriconazole co-therapy were recorded as 0 uniformly.

### 2.2. Tacrolimus and Voriconazole Plasma Concentration Measurement

The tacrolimus blood concentration was determined using a chemiluminescence particle immunoassay. ARCHITECT Tacrolimus Reagent Kit IL77-35 was used for the test. Standard operating procedures for determination, methodology, and stability data are available from the Prograf Assay Kit instruction il77-G08363R10-B1L77C [28].

Automatic two-dimensional liquid chromatography (2D-HPLC, Demeter Instrument Co., Ltd., Changsha, China) was used for voriconazole plasma concentration detection. Chromatographic conditions: column FRO C18 (5 m, 100 mm × 3.0 mm, ANAX), flow rate: 1.0 mL/min, mobile phase: 20 mmol/L ammonium acetate acetonitrile (48:52, *V*/*V*); B column was ASTON HD C18 (150 mm × 4.6 mm, 5 μm, ANAX). D mobile phase was 40 mmol/L ammonium acetate–acetonitrile (85:15, *V*/*V*) at a 1.2 mL/min flow rate. The detection wavelength was 273 nm. The column temperature was 45 °C. The injection volume was 200 μL. The linear range of the method was 0.35–11.26 μg/mL. Annual laboratory quality evaluation was performed through the National Health Commission Clinical Testing Center for all the laboratories.

### 2.3. Genotype and Phenotype Assignment of CYP3A5 and CPY2C19

We used E.Z.N.A SQ. Blood DNA Kit II (OMEGA Bio-Tek, Norcross, GA, USA) reagent for DNA extraction. Subsequently, the Sanger dideoxy DNA sequencing method was used to identify the genotypes of CYP2C19 and CYP3A5. The two samples were detected in Boshang Biotechnology Company (Jinan, China) and Shanghai Aogen Diagnostic Technology Co., Ltd. (Shanghai, China), respectively. The following SNPs were tested: CYP3A5*1(rs15524), CYP3A5*3 (rs776746), CYP2C19*2 (rs4244285), CYP2C19*3 (rs4986893), CYP2C19*17 (rs12248560). Metabolism types were specified according to the genotypes. CYP3A5 metabolism types: extensive metabolizers (EM, CYP3A5*1/*1), intermediate metabolizers (IM, CYP3A5*1/*3), and slow metabolizers (PM, CYP3A5*3/*3) [29].

### 2.4. Statistical Analysis

The Kolmogorov–Smirnov test was used to test the normality of numerical variables. According to their normality, numerical statistical variables were presented as mean ± standard deviation (SD) or median and interquartile range (IQR). *t*-test, Mann–Whitney U, or Kruskal–Wallis H test was chosen according to each applicable condition. For the comparative analysis of classified data, the chi-square test or Fisher’s exact test was used to analyze classified variables. The test level α = 0.05 and bilateral *p* < 0.05 were considered statistically significant. Spearman or point-biserial correlation analysis was used to perform the single-factor analysis. Stepwise multiple linear regression was used to explore predictors of tacrolimus C_Tac_, C/D, and CDW. The inclusion criteria were *p* < 0.05, and the exclusion criteria were *p* > 0.10. A two-sided *p* value < 0.05 was considered statistically significant, and a variance inflation factor (VIF) of >5 was considered indicative of multicollinearity. Statistical analyses were performed using R-4.0.0-WIN, RStudio-1.2.5042, and SPSS software (IBM SPSS Statistics 25, Armonk, NY, USA). Figures were generated using GraphPad Software 8.0 (San Diego, CA, USA) and RStudio-1.2.5042.

## 3. Results

### 3.1. Study Population, TDM Results, and Laboratory Tests

A total of 91 kidney transplant patients were included in this study. The distribution of donor kidney types and genotypes is shown in Table 1. Seventy-three (80.20%) recipients were brain-dead (DBD) donors. Sixty-eight (74.43%) patients were diagnosed with chronic nephritis, and no related etiology was recorded. In comparison, 25 (25.57%) patients were admitted with documented etiology of chronic nephritis. Among them, 11 (12.09%) patients suffered from glomerulonephritis, with 5 (5.49%) diagnosed with IGA nephropathy. The other diagnoses were polycystic disease (5.49%), hypertension (3.30%), diabetes (3.30%), and lupus nephritis (1.10%). The main comorbidities were renal anemia (61.54%), renal hypertension (76.92%), and diabetes (7.69%). Moreover, 25 (27.47%) patients had underlying conditions such as chronic viral hepatitis B, gastritis, enteritis, thyroiditis, heart disease, and dyslipidemia. Voriconazole was used due to suspected or confirmed invasive fungal infection judged by the clinicians. Seven cases (7.70%) were cardiac death (DCD) donors, of which two cases (2.20%) were infantile double kidney donors. Eleven cases (12.10%) were living kidney donors from relatives. Among CYP3A5 genotypes, 6 patients (6.6%) had *1/*1, 46 patients (50.50%) had *1/*3, and 39 patients (42.91%) had *3/*3. Among CYP2C19 genotypes, 35 patients (38.50%) were *1/*1, 40 patients (44.00%) were *1/*2, 12 patients (13.20%) were *1/*3, and 2 patients were *2/*2 and *3/*3, respectively.

Tacrolimus concentration monitoring was performed 1701 times. Among them, 845 (49.67%) concentration points were collected during voriconazole co-therapy. The average daily dose of tacrolimus was 3.33 mg. The average concentration of tacrolimus was monitored 18.7 times per patient. The average blood concentration of tacrolimus C_Tac_ was 7.16 ng/mL, and the median C/D of tacrolimus was 5.59 ng· mL^−1^/mg. The average daily dose of voriconazole was 179.07 mg. Only 1.5% voriconazole was used with a daily dose of more than 400 mg. The medication status and concentration of tacrolimus and voriconazole in patients are also shown in Table 2.

The interquartile ranges of liver function indicators, such as aspartate aminotransferase (AST), alanine aminotransferase (ALT), direct bilirubin (DBIL), and total bilirubin (TBIL), were basically within the normal reference value range, and the median value of creatinine was 170.00 μmol/L. The average hemoglobin level of the study population was 92.39 g/L, and its interquartile range was 87.00 [77.00, 105.00] g/L, indicating that the hemoglobin level of this population was generally low. The statistical description results of the physiological and biochemical indexes of patients are shown in Table 3.

### 3.2. Effect of Voriconazole Co-Therapy on Daily Dose, C_Tac_, C/D, and CDW of Tacrolimus

The daily dose of tacrolimus, C_Tac_, C/D, and CDW was statistically analyzed. Moreover, the analysis results are shown in Table 4.

The daily dose of tacrolimus was significantly lower during voriconazole co-therapy than before. The median daily dose decreased from 5.50 mg to 1.00 mg, a change of nearly 5 times (*p* < 0.0001), as shown in Figure 1A. C_Tac_ did not change significantly (*p* = 0.23), as shown in Figure 1B. While on the contrary, during voriconazole co-therapy, the median value of tacrolimus C/D increased from 1.27 ng·mL^−1^/mg to a median value of 6.96 ng·mL^−1^/mg (*p* < 0.0001; Figure 1C), and similarly the median of CDW increased nearly 5 times (Figure 1D).

### 3.3. Effect of Voriconazole Dose on Daily Dose, C_Tac_, C/D, and CDW of Tacrolimus

We divided the dose of voriconazole into five grades: 0 mg, 100–250 mg, 300–350 mg, 400 mg, and more than 450 mg for further analysis. The effects of voriconazole on the daily dose of tacrolimus, C_Tac_, C/D, and CDW were explored. The analysis results are shown in Table 4 and Figure 2. There were significant differences in tacrolimus daily dose, C_Tac_, C/D, and CDW among different voriconazole levels. When the voriconazole dose was 300–350 mg, the daily dose of tacrolimus was the lowest, the median of C/D was 8.20 ng mL^−1^/mg, and the median value of CDW was 0.15. When comparing them pairwise, these values were significantly higher than those of the 400 mg and >450 mg voriconazole groups.

### 3.4. Effect of Voriconazole Dosage Form on Daily Dose, C_Tac_, C/D, and CDW of Tacrolimus

The main route of voriconazole administration was oral. Moreover, only 136 (15.80%) of the combined voriconazole concentration points were intravenously administrated. We then compared the effect of dosage forms on tacrolimus dose and concentration. The analysis results are also shown in Table 4. There was no significant difference in C_Tac_ between oral and intravenous voriconazole groups (*p* = 0.278). Furthermore, the tacrolimus dose of oral voriconazole patients was lower than the intravenous groups (*p* < 0.0001). The C/D and CDW of oral voriconazole were significantly higher than those of intravenous voriconazole. The median C/D and CDW of oral voriconazole were about 1.5 times higher than the intravenous groups. Hierarchical linear regression was performed to analyze the effects of voriconazole use (Figure 3A,B) and its dosage form (Figure 3C,D) on C/D and CDW in detail. The results showed that the R^2^ of the combined voriconazole group was 0.07, the R^2^ of the non-combined voriconazole group was 0.22, and the R^2^ of the intravenous voriconazole group was 0.19. The R^2^ of the oral voriconazole group was 0.07. The above regression results were statistically significant.

### 3.5. Effect of Voriconazole Concentration on Daily Dose, C_Tac_, C/D, and CDW of Tacrolimus

In this study, the effects of voriconazole concentration on C_Tac_, C/D, and CDW were further explored, and hierarchical linear regression was performed. With the increase in voriconazole concentration, C_Tac_, C/D, and CDW showed an increasing trend (*p* < 0.05). The variation of tacrolimus concentration with voriconazole concentration is shown in Figure 4A. Voriconazole concentration can explain only 0.06% of the variation in C_Tac_, 48.70% of the variation in C/D (Figure 4B), and 49.00% of the variation in CDW. The linear relationship between voriconazole concentration and tacrolimus concentration is weak. Although voriconazole concentration can only explain 0.06% of the tacrolimus concentration change, it still has a strong linear relationship with C/D and CDW of tacrolimus changes. In addition, the effect of voriconazole on C/D and CDW increases with the increase in voriconazole concentration.

### 3.6. Effect of CYP3A5 Genotypes on Daily Dose, C_Tac_, C/D, and CDW of Tacrolimus

According to the CYP3A5 genotype, 91 kidney transplant recipients (1701 concentration points) were divided into three groups: 39 (42.90%) patients with *3/*3 genotype and PM metabolic type, 72 concentration points, accounting for 4.2%. Forty-six (50.50%) cases had *1/*3 genotype and IM metabolic type, and the number of concentration points was 909, accounting for 53.4%. Only six cases had *1/*1 metabolic type and EM, accounting for 6.60%, as shown in Table 5.

In this study, the median C_Tac_ values of *3/*3, *1/*3, and *1/*1 groups were 5.5 ng/mL, 6.6 ng/mL, and 6.55 ng/mL, respectively, and there was no significant difference in C_Tac_ distribution among the three groups (*p* = 0.237). The median C/D of the *1/*1 group was 1.88, and the median C/D values of *1/*3 and *3/*3 groups were 1.80 and 5.18, respectively. The statistical results were significantly different (*p* < 0.001). The results of CDW analysis were the same as those of C/D, showing similar distribution characteristics among the three groups (Figure 5). C/D and CDW of patients with CYP3A5 genotype *3/*3 were nearly 3 times higher than those of patients with genotype *1/*1 or *1/*3. Therefore, patients with CYP3A5 genotype *3/*3 need to pay more attention to the change in tacrolimus concentration when combined with drugs. CYP3A5 genotype *3/*3 is still statistically different from *1/*3 and *1/*1 groups (see Figure 5). Stratified regression of C_Tac_, C/D, and CDW in different CYP3A5 genotype groups is shown in Figure 4E,F. The R^2^ of CYP3A5 *3/*3 genotype groups was the lowest, showing that the influence of CYP3A5 *3/*3 was responsible for the greatest variation in tacrolimus concentration after kidney transplantation.

### 3.7. Effect of CYP3A5 Genotypes on Daily Dose, C_Tac_, C/D, and CDW of Tacrolimus

According to the CYP2C19 genotype, 91 kidney transplant recipients (1701 concentration points) were divided into five groups (see Table 5 for details). In this study, the median daily doses of tacrolimus for CYP2C19 genotypes 1/*1, *1*2, *1/*3, *2/*2, and *2/* were 3.00, 3.50, 3.00, 0.25, and 0.5 mg, respectively. The median concentrations of tacrolimus were 6.80, 6.50, 5.80, 7.65, and 3.47 ng· mL^−1^, respectively. There were significant differences in dose and C_Tac_ distribution (*p* < 0.001). C/D and CDW analysis results show the following: The C/D and CDW of the *2/*2 group were significantly higher than those of the other groups, the median C/D was 22.11 ng· mL^−1^/mg, and the CDW was 0.37 ng·mL^−1^·mg^−1^/kg. The results of the *2/*3 group were also higher than those of the other genotype groups except the *2/*2 group (*p* < 0.001), suggesting that patients with genotypes *2/*2 and 2/*3 may cause higher fluctuations in tacrolimus concentration than patients with other genotypes (Figure 6).

### 3.8. Determinants of Tacrolimus C_Tac_, C/D, and CDW

Further stepwise multiple linear regression of C/D found that compared with patients with CYP3A5 genotype *1*3, the C/D value of tacrolimus increased by 1.239 (*p* < 0.001) in patients with CYP3A5*3*3. Furthermore, the C/D value of tacrolimus increased by 2.830 when voriconazole concentration increased by 1.0 μg/mL (*p* < 0.001). Physiological and biochemical indexes, including lymphocyte count, blood urea nitrogen, and serum creatinine (CREA), can also induce the change in C/D, but the correlation effect intensity was weak. Tacrolimus C/D decreased by 0.854 (*p* = 0.004) after the co-administration of drugs metabolized by CYP2C19, while the C/D decreased by 1.314 after the co-administration of CYP2C19 inducers. The VIF values of this model were all below 5, without multicollinearity, and the F statistic was 106.3 (*p* < 0.001). The adjusted R square of this model was 0.5479, indicating that the above-combined predictors could explain 54.79% of the C/D changes of tacrolimus. Predictors of voriconazole trough concentration are presented in Table 6.

The final multiple linear regression equation is also shown as follows:C/D = 5.893 + 1.491 × sex × A + 1.239 × CYP3A5 genotype × B − 0.911 × Tac Daily dose − 0.041 × Postoperative time − 0.006 × VRC Daily Dose + 2.830 × C_VRC_ − 0.043 × Lymphocyte count + 0.044 × Blood urea nitrogen − 0.002 × Serum creatinine − 0.854 × CYP2C19 substrate × C − 1.314 × CYP2C19 inducer × D(1)

“A = 1” if the sex of the patient is male, otherwise “A = 0”; “B = 1” if the genotype of CYP3A5 is *3*3, otherwise “B = 0”; “C = 1” if the patient has concomitant drug use of CYP2C19 substrate, otherwise “C = 0”; “D = 1” if the patient has concomitant drug use of CYP2C19 inducer, otherwise “D = 0”.

Then we performed multiple linear regression analyses of tacrolimus C_Tac_ and CDW. The results are shown in Appendix A. The adjusted R square of the C_Tac_ model was only 0.1676 (F = 10.29; *p* < 0.001). The adjusted R square of the CDW model was 0.5161 (*p* < 0.001), which is close to the result of C/D.

## 4. Discussion

In this study, we systematically analyzed the effect of voriconazole on tacrolimus dose and concentration. Results showed that the median tacrolimus dose might decrease by 5.5 times with voriconazole co-therapy. On the contrary, the median of C/D and CDW increased dramatically by 6 times, accompanied by significant interindividual differences, consistent with some previous findings [31,32,33]. Studies have also found that intravenous injection and oral combination of voriconazole increased the C/D variation by 642.1% and 994.1%, respectively. Moreover, the C/D of tacrolimus patients in the oral administration group was 1.54 times higher than those in the intravenous administration group. As for the influence of the CYP3A5 genotype, the Kruskal–Wallis test and multiple linear regression of C_Tac_, C/D, and CDW all showed that for the *3*3 CYP3A5 genotype, C_Tac_, C/D, and CDW were higher than those of patients with other genotypes, consistent with existing evidence [22,34].

Meanwhile, the influence of CYP2C19 genotype, C/D, and CDW in CYP2C19*2/*2 and 2/*3 groups were significantly higher than those in other genotype groups, which is similar to the study results of Vanhove [24]. The combined clinical predictors explained only 16.76% of the variation in C_Tac_, while the prediction efficiency of C/D and CDW was more than 50%. This difference is most likely because tacrolimus dosing must be constantly adjusted to achieve a uniform target concentration during treatment, resulting in a minimal difference in concentrations between patients. It is likely that the relationship between multiple factors (such as voriconazole co-therapy, dosage form, and CYP3A5 genotype) and C_Tac_ in this study was not statistically significant. Similar findings have been reported by other authors [24,35]. We further compared the predictors and predictive ability of several previous models. The results are shown in Table 7.

Rangel et al. found that the long-term survival of pancreas allografts was affected not only by rejection but also by immunosuppressive regimen toxicity such as metabolic disorders or infection [40]. The conclusion was also applicable for patients after renal transplantation [41,42,43,44,45]. Therefore, the results of our study may not only reduce rejection caused by drastic fluctuations in tacrolimus concentrations, but also reduce other immunosuppressive diseases. It may also help clinicians using tacrolimus more precisely, minimizing its toxic effects and finally resulting in prolonged survival after transplantation. In addition, it is the first study to include voriconazole concentration as a predictor in the final model instead of whether voriconazole was used. Moreover, the model in this study has the best predictive performance compared with other similar studies included in Table 7. This indicated that the effect of voriconazole on tacrolimus should be quantified by voriconazole concentration, which was more scientific and reasonable. Consistent with existing work [46,47,48], we also found that CYP3A5*3/*3 carriers required a lower dose of tacrolimus and presented a higher level of C/D. However, although the effects of voriconazole on C_Tac_, C/D, or CDW were systematically analyzed in this study, and the C/D and CDW prediction models with good prediction efficiency were established, there were still some limitations in this study. Firstly, only CYP3A5 and CYP2C19 genotypes were analyzed in this study, and the influence of other genotypes, such as CYP3A4*18B, CYP3A4*22, SLCO1B1, and ABCB1, was not considered [22,39]. Stephania et al. found that MDR1 2677 C/C was associated with higher tacrolimus trough levels, while MDR1 1236 T/T was associated with lower tacrolimus levels and higher doses [46]. In their study, they also found that POR*28 rs1057868 1508C > T exhibited no associations with C/D. Lunde et al. found that patients with POR*28 and PPARA variant alleles demonstrated 15% lower (*p* = 0.04) and 19% higher (*p* = 0.01) tacrolimus C/D, respectively [48]. In addition, the results of Li et al. suggested that genetic polymorphisms of CYP3A4*18B may be partly responsible for the large interindividual variability of tacrolimus blood levels in Chinese renal transplant patients [47]. Therefore, the functional consequence of the variant is still unclear and divergent and still needs further study. In addition, fewer patients with CYP2C19 genotypes *2*2 and *2*3 and CYP2C19*17 are rarely found in Asian populations. Patients with CYP2C19 SNP *17 were not included in this study. Secondly, this study only studied the effect of voriconazole on the most common oral dosage form of tacrolimus, excluding intravenous dosage forms and sustained-release preparations. Thirdly, considering the aftereffects of voriconazole metabolism, the change in tacrolimus concentration after voriconazole discontinuation was not explored in this study. Prospective studies should be conducted to further analyze its effect on tacrolimus. Moreover, although drug interactions were included in the final equation, drug interactions are also more complex during practical clinical applications. The effect of potential drug interactions still deserves further investigation. In addition, this study was carried out based on real-world data without intervention in clinical treatment, and most of the data came from the HIS. Therefore, bias in medical order recording should be considered a confounding factor. Above all, we noticed that several institutions had conducted population pharmacokinetic studies on liver or lung transplant recipients and hematopoietic stem cell transplantation patients. Based on influencing factors and influencing intensity, a population pharmacokinetic study on kidney transplantation patients is needed to recommend the dose of tacrolimus under different voriconazole concentrations.

## 5. Conclusions

Voriconazole was the main factor causing C/D and CDW variation, and the effect intensity should be quantitative by its concentration. For kidney transplant recipients with CYP3A5 genotype of *3/*3 and CYP2C19 genotype of *2/*2 and *2/*3, more attention should be given to the tacrolimus concentration change during voriconazole co-therapy. The influence factor model established in this study could explain more than 50% of C/D and CDW variation. It may help to predict the tacrolimus plasma concentration of kidney transplant recipients, reduce the occurrence of rejection, and improve the efficacy of transplantation.

## Figures and Tables

**Figure 1 pharmaceutics-14-02739-f001:**
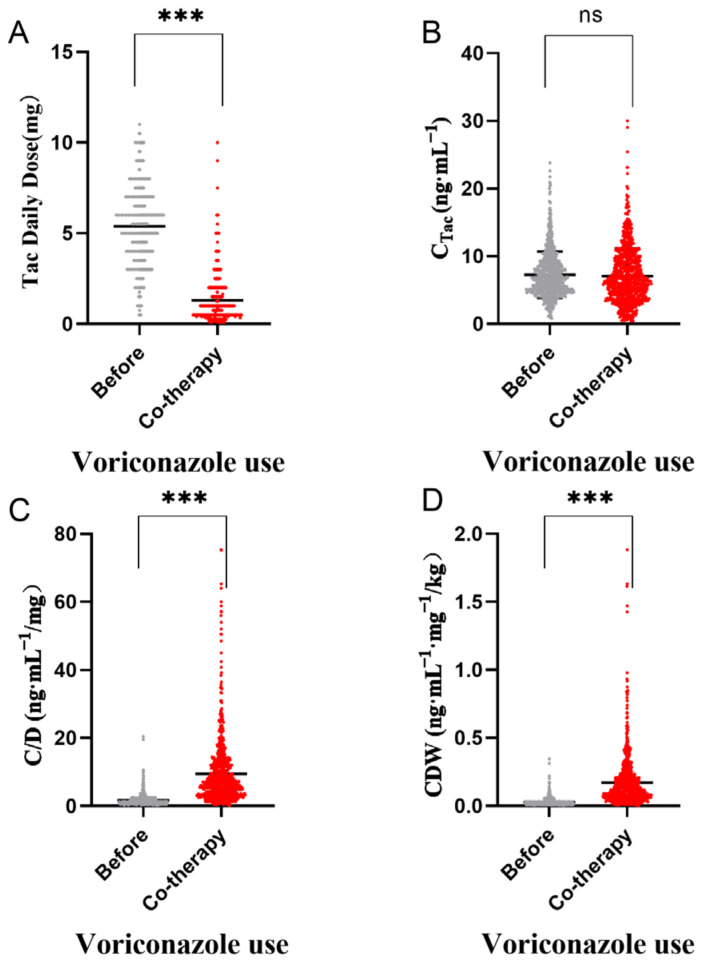
Effects of voriconazole use on tacrolimus daily dose (**A**), C_Tac_ (**B**), C/D (**C**), and CDW (**D**). *** *p* < 0.001; *p* value was tested by Kruskal–Wallis test and adjusted by Dunn–Bonferroni correction; direct concentration (C_Tac_), standard concentration adjusted by tacrolimus daily dose (C/D), and weight-adjusted standard concentration (CDW).

**Figure 2 pharmaceutics-14-02739-f002:**
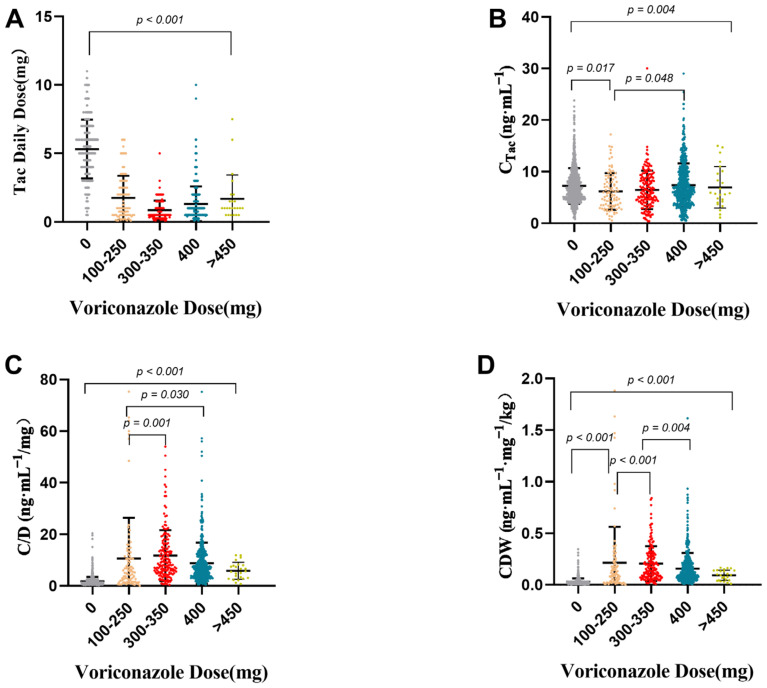
Effects of voriconazole dose on tacrolimus daily dose (**A**), C_Tac_ (**B**), C/D (**C**), and CDW (**D**). p value was tested by Kruskal–Wallis test and adjusted by Dunn–Bonferroni correction; direct concentration (C_Tac_), standard concentration adjusted by tacrolimus daily dose (C/D), and weight-adjusted standard concentration (CDW).

**Figure 3 pharmaceutics-14-02739-f003:**
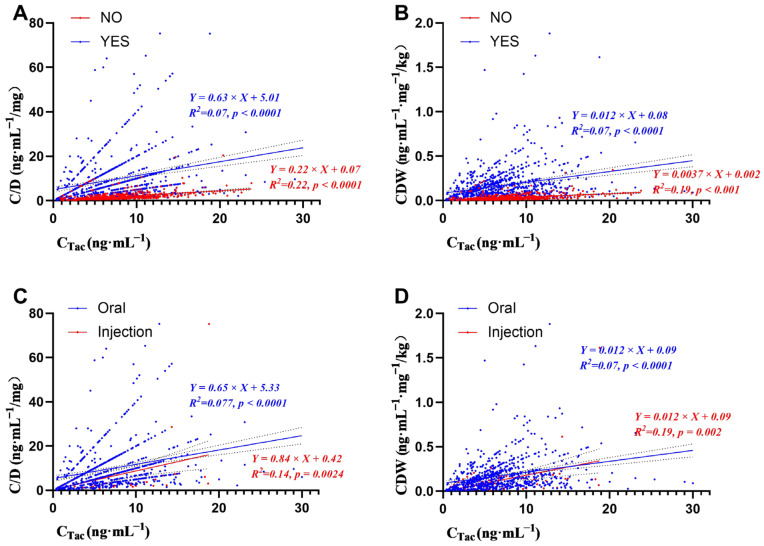
Stratified linear regression of voriconazole use (**A**,**B**) and its dosage form (**C**,**D**) on C/D (**A**,**C**) or CDW (**B**,**D**). Blue line and dots represent the regression line and values during voriconazole co-therapy; while red line and dots represent the regression line and values without voriconazole co-therapy.

**Figure 4 pharmaceutics-14-02739-f004:**
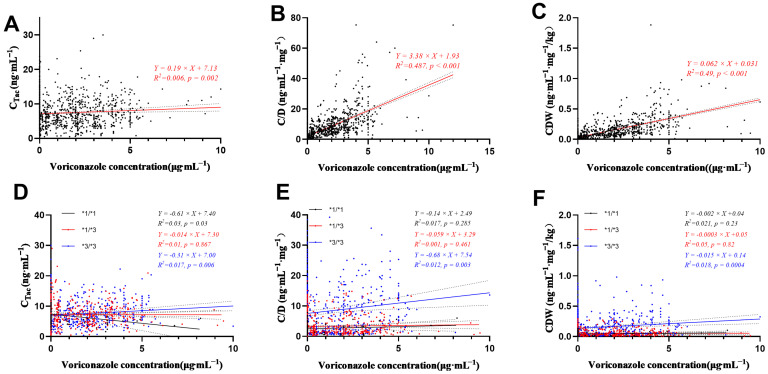
Linear regression of voriconazole concentration with C_Tac_ (**A**), C/D (**B**), and CDW (**C**) and stratified regression of voriconazole concentration (C_VRC_) with C_Tac_ (**D**), C/D (**E**), and CDW (**F**) in different CYP3A5 genotype groups. In (**D**–**F**), black line and dots represent the regression line and values in groups with genotype of CYP3A5*1/*1; blue line and dots represent the regression line and values in groups with genotype of CYP3A5*3/*3; red line and dots represent the regression line and values in groups with genotype of CYP3A5*1/*3.

**Figure 5 pharmaceutics-14-02739-f005:**
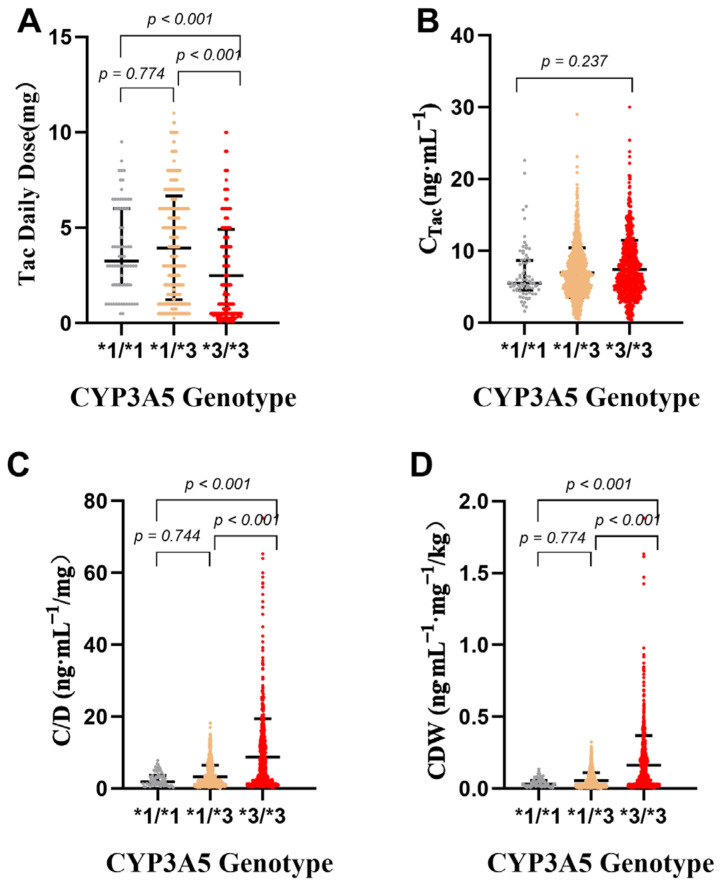
Effects of CYP3A5 genotypes on tacrolimus daily dose (**A**), C_Tac_ (**B**), C/D (**C**), and CDW (**D**). *p* value was tested by Kruskal–Wallis test and adjusted by Dunn–Bonferroni correction; direct concentration (C_Tac_), standard concentration adjusted by tacrolimus daily dose (C/D), and weight-adjusted standard concentration (CDW).

**Figure 6 pharmaceutics-14-02739-f006:**
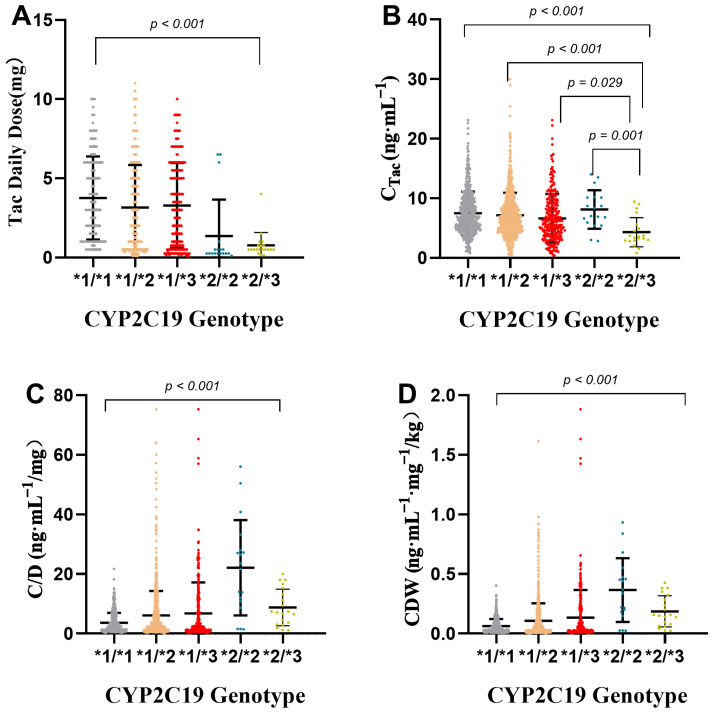
Effects of CYP2C19 genotypes on tacrolimus daily dose (**A**), C_Tac_ (**B**), C/D (**C**), and CDW (**D**). *p* value was tested by Kruskal–Wallis test and adjusted by Dunn–Bonferroni correction; direct concentration (C_Tac_), standard concentration adjusted by tacrolimus daily dose (C/D), and weight-adjusted standard concentration (CDW).

**Table 1 pharmaceutics-14-02739-t001:** Distribution of kidney sources and genotypes of patients.

Characteristics (*N* = 91)	Value
Male, n (%)	70 (76.90)
Age (years) ^a^	40.00 [32.00, 49.00]
Wt (kg) ^b^	60.02 (13.14)
Etiology of chronic nephritis	
Unknown cause	68 (74.43)
Glomerulonephritis	11 (12.09)
Polycystic disease	5(5.49)
Hypertension	3 (3.30)
Diabetes	3 (3.30)
Lupus nephritis	1 (1.10)
Kidney source, n (%)	
DBD	73 (80.2)
DCD	7 (7.70)
RD	11 (12.10)
Genotype of CYP3A5, n (%)	
*1/*1	6 (6.60)
*1/*3	46 (50.50)
*3/*3	39 (42.90)
Genotype of CYP2C19, n (%)	
*1/*1	35 (38.50)
*1/*2	40 (44.00)
*1/*3	12 (13.20)
*2/*2	2 (2.20)
*2/*3	2 (2.20)

^a^ median (IQR); ^b^ mean ± SD; DBC: brain-dead organ donation; DCD: cardiac death organ donation; RD: relative organ donation.

**Table 2 pharmaceutics-14-02739-t002:** Drug use and concentration information of patients.

Parameter	Value
Number of C_Tac_	1701
Tac daily dose (mg) ^b^	3.33 ± 2.68
C_Tac_ (ng·mL^−1^) ^b^	7.16 ± 3.77
C/D (ng·mL^−1^/mg) ^a^	6.50 [4.60, 9.10]
CDW (ng·mL^−1^·mg^−1^/kg) ^a^	0.04 [0.02, 0.12]
No. of C_VRC_	1455(85.54%)
C_VRC_ (μg·mL^−1^) ^a^	0.00 [0.00, 1.98]
VRC daily dose (mg), n (%)	
0	845 (50.17)
100	9 (0.53)
150	8 (0.47)
200	72 (4.23)
250	3 (0.18)
300	140 (8.23)
350	27 (1.59)
400	558 (32.69)
450	11 (0.65)
500	1 (0.06)
600	9 (0.53)
800	3 (0.18)

^a^ median (IQR); ^b^ mean ± SD, C/D = C_Tac_ (ng·mL^−1^) / daily dose (mg); CDW = C_Tac_ (ng·mL^−1^) / [daily dose (mg) / weight(kg)].

**Table 3 pharmaceutics-14-02739-t003:** Laboratory parameters.

Parameter	Value
White blood cell count (10^9^/L) ^a^	7.07 [5.16, 9.45]
Red blood cell count (10^12^/L) ^a^	2.98 [2.57, 3.68]
Lymphocyte count (%) ^b^	14.07 ± 11.48
Neutrophilic granulocyte (%) ^b^	78.41 ± 14.68
Hematocrit (%) ^a^	26.70 [23.50, 32.00]
Hemoglobin (g/L) ^a^	87.00 [77.00, 105.00]
Blood platelet count (10^9^/L)	176.36 ± 70.69
Alanine aminotransferase (U/L) ^a^	12.90 [8.05, 21.70]
Aspartate aminotransferase (U/L) ^a^	13.90 [10.50, 20.30]
Total bilirubin (μmol/L) ^a^	6.70 [5.00, 8.80]
Direct bilirubin (μmol/L) ^a^	2.70 [1.95, 3.70]
Total bile acid (μmol/L) ^a^	3.50 [2.30, 5.69]
Albumin (g/L) ^b^	34.42 ± 4.42
Blood urea nitrogen (mmol/L) ^a^	16.20 [10.73, 26.64]
Serum creatinine (μmol/L) ^a^	170.00 [120.20, 306.05]
CCR * (mL/min) ^a^	40.72 [22.97, 60.01]
Prothrombin time (s) ^a^	12.80 [11.80, 13.70]
International normalized ratio ^a^	1.04 [0.93, 1.11]
Prothrombin activity (%) ^a^	97.50 [89.00, 110.80]
Procalcitonin (ng/mL) ^a^	0.17 [0.09, 0.39]
C reactive protein (mg/L) ^a^	5.56 [1.75, 28.05]

^a^ median (IQR); ^b^ mean ± SD; * creatinine clearance rate: the calculation of kidney clearance rate (CCR) was based on the calculation formula [30]; CCR (male) = [(140 −age) × weight (kg)] / [0.818 × CREA(μmol/L)]; CCR(female) = [(140 − age) × weight (kg)] / [0.818 × CREA(μmol/L)] × 0.85.

**Table 4 pharmaceutics-14-02739-t004:** Effects of voriconazole use, dose, and dosage form on daily dose, CTac, C/D, and CDW of tacrolimus.

Parameter		Sample (N, %) ^a^	C_Tac_ ^b^ (ng·mL^−1^)	Daily Dose ^b^ (mg)	C/D ^b^ (ng mL^−1^/mg)	CDW ^b^ (ng·mL^−1^·mg^−1^/kg)
Voriconazole use	Yes	84 (49.67)	7.05 ± 4.09	1.00 (0.5–2.00)	6.96 (3.62–12.00)	0.12 [0.06, 0.21]
	No	856 (50.33)	7.27 ± 3.41	5.50 (4.00–7.00)	1.27 (0.90–1.83)	0.02 [0.01, 0.03]
	*p*	/	0.23	<0.0001	<0.0001	<0.001
Voriconazole Dosage form	Injection	66 (7.96)	1.00 [0.50, 3.00]	6.75 [4.70, 10.00]	5.00 [2.60, 7.35]	0.09 [0.05, 0.15]
	Oral	763 (92.04)	1.00 [0.50, 1.50]	6.20 [4.10, 9.25]	7.20 [3.98, 12.60]	0.12 [0.07, 0.22]
	*P*	/	<0.001	0.278	<0.001	0.007
Voriconazole daily dose (mg)	0	860 (50.56)	6.60 [4.90, 8.90]	5.50 [4.00, 7.00]	1.27 [0.91, 1.84]	0.02 [0.01, 0.03]
	100–250	92 (5.41)	5.50 [3.40, 8.50]	1.00 [0.50, 2.50]	5.31 [1.75, 11.50]	0.10 [0.03, 0.22]
	300–350	167 (9.82)	5.90 [4.10, 8.75]	0.50 [0.40, 1.25]	8.20 [5.27, 15.10]	0.15 [0.08, 0.29]
	400	558 (32.80)	6.55 [4.40, 9.80]	1.00 [0.50, 2.00]	7.00 [3.63, 11.80]	0.12 [0.07, 0.20]
	>450	24 (1.41)	5.75 [4.15, 9.72]	1.00 [0.88, 1.62]	5.60 [3.00, 7.85]	0.09 [0.04, 0.14]
	*p*	<0.001	0.004	<0.001	<0.001	<0.001

Data are represented as median (IQR); ^a^ Pearson chi-square test; ^b^ Kruskal–Wallis test.

**Table 5 pharmaceutics-14-02739-t005:** Effects of CYP3A5 and CYP2C19 on daily dose, C_Tac_, C/D, and CDW of tacrolimus.

Parameter		Sample (N, %) ^a^	C_Tac_ ^b^ (ng·mL^−1^)	Daily Dose ^b^ (mg)	C/D ^b^ (ng mL^−1^/mg)	CDW ^b^ (ng·mL^−1^·mg^−1^/kg)
CYP3A5	EM (*1/*1)	6 (6.60)	5.50 [4.57, 8.62]	3.25 [2.00, 6.00]	1.88 [1.28, 3.60]	0.03 [0.02, 0.06]
	IM (*1/*3)	46 (50.50)	6.60 [4.70, 8.90]	3.50 [1.50, 6.00]	1.80 [1.09, 4.70]	0.03 [0.02, 0.08]
	PM (*3/*3)	39 (42.90)	6.55 [4.60, 9.70]	1.25 [0.50, 4.00]	5.18 [1.50, 12.29]	0.09 [0.03, 0.22]
	*p*	<0.001	0.237	<0.001	<0.001	<0.001
CYP2C19	*1/*1	35 (38.50)	6.80 [5.00, 9.50]	3.00 [1.00, 6.00]	2.20 [1.20, 5.40]	0.04 [0.02, 0.09]
	*1/*2	40 (44.00)	6.50 [4.70, 8.90]	2.50 [0.50, 5.00]	2.92 [1.24, 8.00]	0.05 [0.02, 0.14]
	*1/*3	12 (13.20)	5.80 [3.70, 8.80]	3.00 [0.75, 5.50]	2.03 [1.00, 9.00]	0.04 [0.02, 0.19]
	*2/*2	2 (2.20)	7.65 [6.08, 10.10]	0.25 [0.25, 0.50]	22.60 [11.10, 27.80]	0.35 [0.19, 0.47]
	*2/*3	2 (2.20)	3.47 [2.90, 5.08]	0.50 [0.50, 0.75]	7.25 [3.55, 13.80]	0.15 [0.08, 0.29]
	*p*	<0.001	<0.001	<0.001	<0.001	<0.001

^a^ Pearson chi-square test; ^b^ Kruskal–Wallis test; data are represented as median (IQR).

**Table 6 pharmaceutics-14-02739-t006:** The independent influencing factors of C/D.

Parameters	Estimate Coefficients	Std. Error	*t*	VIF	*p*
(Intercept)	5.893	0.729	8.080		<0.001
Male	1.491	0.384	3.881	1.147	<0.001
CYP3A5*3*3	1.239	0.337	3.676		<0.001
Tac Daily Dose	−0.911	0.085	−10.671	2.449	<0.001
Postoperative time	−0.041	0.017	−2.405	1.281	0.016
VRC Daily Dose	−0.006	0.001	−4.719-	3.031	<0.001
C_VRC_	2.830	0.156	18.127	2.336	<0.001
Lymphocyte count	−0.043	0.015	−2.870	1.312	0.004
Blood urea nitrogen	0.044	0.014	3.046	2.016	0.002
Serum creatinine	−0.002	0.001	−2.167	1.990	0.030
CYP2C19 substrate ^a^	−0.854	0.297	2.879	1.071	0.004
CYP2C19 inducer ^b^	−1.314	0.587	−2.238	1.063	0.025
F R^2^ Adjusted R^2^		106.3			
		0.5479			
		0.5427			
*p*		<0.001			

Multiple linear regression was performed using the stepwise method. The standard for inclusion was 0.05; while the standard for exclusion was 0.10 (*N* = 1701). ^a^ the patient has concomitant drug use of CYP2C19 substrate; ^b^ the patient has concomitant drug use of CYP2C19 inducer.

**Table 7 pharmaceutics-14-02739-t007:** Predictors associated with tacrolimus concentration in different studies.

Reference	Transplantation Type	No.	DV ^a^	Predictors	Method	R^2^
Our study	Kidney	91	C/D	CYP3A5 genotype, POT, Tac Daily dose, VRC Daily dose, C_VRC_, LYM, BUN CREA, CYP2C19 substrate, and CYP2C19 inducer	Multiple linear regression	0.548
Vanhove T. et al. [24]	Kidney and lung	126	C/D	HCT, age, CYP3A5 genotype, and CYP3A4 substrate	Multiple linear regression	0.22
Janaína F. et al. [35]	Kidney	127	C/D	Age, POT, CYP3A5, and PPARA genotype	Multiple linear regression	0.123
Pinon M. et al. [36]	Liver	49	CDW	CYP3A5 genotype and GRWR	Multiple linear regression	0.173
Chengxin L. et al. [37]	Hematopoietic stem cell	46	C/D	Sex, weight, POT, HGB, PLT, CREA, and VRC	Multiple linear regression	0.33
Lizhi C. et al. [38]	Kidney	142	C	CREA, HCT, Wu-zhi capsule, CYP3A5 genotype, and Tac daily dose	Correlation analysis	NA
Suetsugu. K. et al. [22]	Hematopoietic stem cell	36	C/D ^a^	CYP3A4 genotype, POR*1/*1 CYP2C19 genotype, and VRC	Multiple logistical regression	NA
Yi W. et al. [39]	Liver	210	C/D	Tac daily dose, POT, TBIL, and SLCO1B1 rs2291075	Multiple factor analysis	NA

^a^ DV: the dependent variable is the increase in C/D; NA: not applicable or not mentioned. POT, postoperative time; BUN, blood urea nitrogen; CREA, creatinine; GRWR, the graft-to-recipient weight ratio; HCT, hematocrit; HGB, hemoglobin; LYM, lymphocyte; PLT, platelet count; PPARA, peroxisome proliferator-activated receptor alpha; POR, cytochrome P450 oxidoreductase; TBIL, total bilirubin; SLCO1B1, solute carrier organic anion transporter family member 1B1; VRC, voriconazole.

## Data Availability

We declare that our research data are available on reasonable request. The data will include (but are not limited to) raw data, processed data, software, and algorithms. Researchers can contact us if they need the research data.

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
