# Peer review of "The Effect of Voriconazole on Tacrolimus in Kidney Transplantation Recipients: A Real-World Study"

_pharmaceutics, 2022, doi:10.3390/pharmaceutics14122739_

Round 1
Reviewer 1 Report
As I understood, there was only one group in this study. However, in Line 162 the Authors mention "the non-voriconazole co-therapy group." I guess the Authors mean patients before the start of co-therapy, or after its discontinuation. This should be corrected.
Similarly, in Tab. 2, it is stated that in approx. 50% of measurements, the daily dose of voriconazole was 0. What does it mean? Were the measurements done before co-therapy, or after its discontinuation? I think this issue is important, as it may influence the results (e.g. pharmacokinetics on the 1-st day after the discontinuation of co-therapy differs from the pharmacokinetics before its start). This should be clarified.
Does the Tab. 3 present the results BEFORE the start of co-therapy, or DURING it? I think results before, during, and after the co-therapy should not be mixed together.
Reviewer 2 Report
In the manuscript entitled “The effect of voriconazole on tacrolimus in kidney transplantation recipients: a real-world study”, the authors evaluated a prediction model of tacrolimus daily dose and weight-adjusted standard concentration in kidney transplant recipients under voriconazole co-therapy. Different voriconazole concentrations were used as predictors in the final model. They also verified the effect of the genotype of CYP3A5 and CYP2C19 on those parameters. The study is interesting and can help clinicians during their practice. Before we proceed, please address the following comments.
Minor comments
1) Provide information about the transplant, such as the time of the transplant, the etiology of the chronic kidney disease, and comorbidities. Likewise, what was the indication for the voriconazole prescription?
2) Please acknowledge the use of other drugs that can impact tacrolimus trough levels: increase (amlodipine/verapamil/diltiazem, clarithromycin/erythromycin/azithromycin, protease inhibitors, methylprednisolone) or decrease (rifampicin, isoniazid, carbamazepine, trimethoprim, imipenem, ciprofloxacin, and cephalosporin) (reference: Alakhali et al, JPCS Vol 8, Jan-March 2014).
3) In Table 3, is fasting blood glucose available?
4) As tacrolimus has a narrow therapeutic window and toxicity, it is important to discuss the importance of the findings not only for rejection, but also for renal toxicity, metabolic disorders, infection, etc (doi: 10.1517/17425255.2014.964205).
5) In addition, discuss the effect of other genetic polymorphisms in kidney transplant recipients. Such as MDR1 and POR (doi: 10.3389/fphar.2021.674117).
